# SMOS Third Mission Reprocessing after 10 Years in Orbit

**Roger Oliva [1],\*, Manuel Martín-Neira [2] , Ignasi Corbella [3] , Josep Closa [4], Albert Zurita [4], François Cabot [5], Ali Khazaal [5], Philippe Richaume [5], Juha Kainulainen [6], Jose Barbosa [7], Gonçalo Lopes [8], Joseph Tenerelli [9], Raul Díez-García [10], Veronica González-Gambau [11] and Raffaele Crapolicchio [12]**

1.  Zenithal Blue Technologies S.L.U., 08023 Barcelona, Spain
2.  European Space Research and Technology Centre, European Space Agency, 2201 AZ Noordwijk, The Netherlands; Manuel.Martin-Neira@esa.int
3.  Remote Sensing Laboratory, Universitat Politecnica de Catalunya, 08034 Barcelona, Spain; corbella@tsc.upc.edu
4.  Airbus Defence and Space, Microwave Instruments department, 28692 Madrid, Spain; Josep.Closa@airbus.com (J.C.); alberto.Zurita@airbus.com (A.Z.);
5.  Centre D'Etudes Spatiales de la Boisphere, 31400 Toulouse, France; francois.cabot@cesbio.cnes.fr (F.C.); Ali.Khazaal@cesbio.cnes.fr (A.K.); philippe.richaume@cesbio.cnes.fr (P.R.)
6.  Harp Technologies, 02150 Espoo, Finland; juha.kainulainen@harptechnologies.com
7.  Research and Development in Aerospace, 8006 Zurich, Switzerland; jose.barbosa@rdaerospace.ch
8.  Deimos Engenheria, 1998-023 Lisbon, Portugal; goncalo.lopes@deimos.com.pt
9.  OceanDataLab, 29280 Locmaria Plouzané, France; joseph.tenerelli@oceandatalab.com
10.  European Space Astronomy Centre, European Space Agency, 28692 Madrid, Spain; raul.diez.garcia@esa.int
11.  Department of Physical Oceanography, Institute of Marine Sciences, CSIC and Barcelona Expertise Center, 08003 Barcelona, Spain; vgonzalez@icm.csic.es
12.  European Space Research Institute, European Space Agency, 00044 Frascati, Italy; raffaele.crapolicchio@esa.int
*   Correspondence: r.oliva@zenithalblue.com

**Abstract:** After more than 10 years in orbit, the SMOS team has started a new reprocessing campaign for the SMOS measurements, which includes the changes in calibration and image reconstruction that have been made to the Level 1 Operational Processor (L1OP) during the past few years. The current $L_1$ processor, version v620, was used for the second mission reprocessing in 2014. The new version, v724, is the one run in the third mission reprocessing and will become the new operational processor. The present paper explains the major changes applied and analyses the quality of the data with different metrics. The results have been obtained with numerous individual tests that have confirmed the benefits of the evolutions and an end-to-end processing campaign involving three years of data used to assess the improvements of the SMOS measurements quantitatively.

**Keywords:** SMOS; calibration; radiometry; reprocessing

## 1. Introduction

The Soil Moisture and Ocean Salinity (SMOS) mission is the second Earth Explorer mission of the European Space Agency (ESA). The satellite was launched in November 2009 and has been continuously operating ever since, with an excellent health status. Data acquisition is in the order of 99.88%, and processing performance is above 99%. As such, the ESA has continuously provided nominal and near-real-time data for the past 10 years since the end of the commissioning phase.

The original objectives of soil moisture [1] and sea surface salinity [2] have been complemented with new applications, such as to thin sea-ice thickness, severe winds over ocean and freeze/thaw soil state products [3]. The satellite contains a single payload, the MIRAS (Microwave Imaging Radiometer using Aperture Synthesis), the first ever space-based L-band interferometric radiometer [4]. Even though interferometric radiometers have long been used by radio-astronomers, having such an instrument space-based for earth observation missions has presented several challenges. More than ten years after launch, the SMOS team continues to improve the calibration and the image reconstruction processes. As a result of this, new processor versions are developed, and when the changes in quality are considered important, the SMOS team prepares for a new reprocessing. Currently, SMOS is preparing the third mission reprocessing with the L1OP v724. The Methods section provides an overview of the changes involved in the new version with respect to the v620 operational version used in the second mission reprocessing. The Results section assesses the end-to-end improvements of the data.

## 2. Methods

In this section, we present the improvements that were applied to the v620 processing baseline to form the new v724 processing baseline. A high-level overview of the SMOS Level 1 processing baseline is presented in Figure 1.

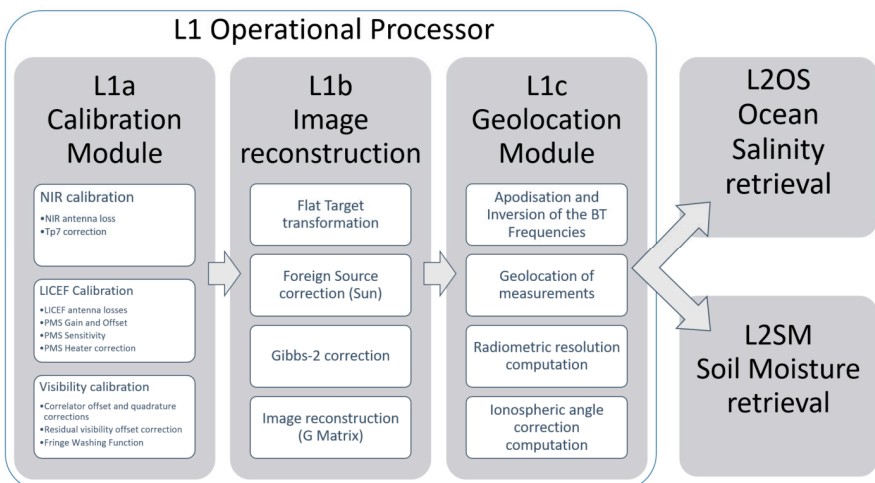

**Figure 1.** High-level architecture of the SMOS Operational processing baseline.

The improvements in the new L1OP were applied to the calibration process and the image reconstruction processing. The improvements for these two categories are presented below in separate sub-sections.

### 2.1. Changes in Calibration

Calibration is a process where raw MIRAS data, including, e.g., cross-correlations between all the pairs formed by the 72 MIRAS receivers, counts of the receivers' total power detectors, and noise injection radiometer pulse lengths, are turned into radiometric observables like antenna temperatures and power levels. The calibration of the MIRAS instrument as a whole is a complex process including several steps. An overview of the calibration process can be found, e.g., in [5].

For the v724 processing baseline, five main improvements were done in this calibration process. They are improvements related to the following:

- Noise injection radiometer (NIR) calibration strategy.
- NIR antenna losses.
- Power measurement system (PMS) sensitivity factors.
- PMS heater correction.

-  Thermal latency of the temperature sensor in NIR antennas.

In the following sub-sections, we describe these updates in detail.

### 2.1.1. NIR Calibration

Analysis of the second mission reprocessing led to the following conclusion: the calibration of the NIR parameters, the noise injection temperature (Tna) and the level of the noise injection (Tnr) [6] were introducing a bias in the stability of the measurements. This was evident when looking at the bias of the measurements over a large portion of the Pacific open ocean with respect to the ocean forward model [7]. The comparison of those biases showed a large negative correlation with the variation in the main NIR calibration parameter, Tna. Figure 2 shows such a comparison for X polarisation measurements in ascending orbits.

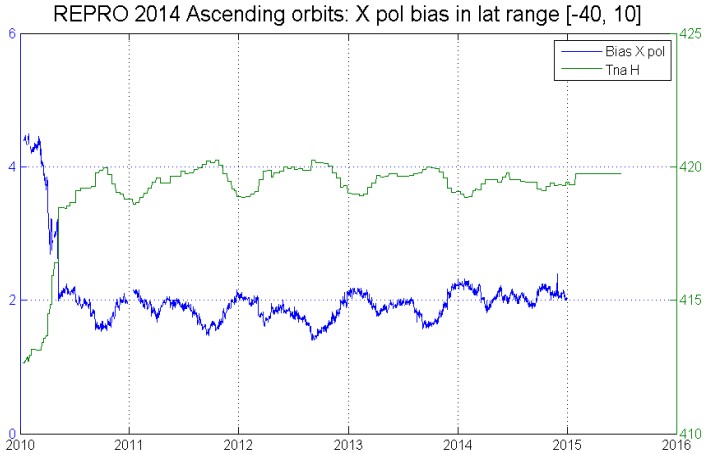

**Figure 2.** Bias of the Xpol SMOS measurements for ascending orbits over the Pacific Ocean (blue) for the period 2010–2015 and the NIR calibration parameter Tna (green).

A computation of the correlation factor between the two variables is provided in Table 1.

**Table 1.** Correlation factor between brightness temperature bias as computed over the ocean and NIR calibration parameter Tna.

| Polarization | Ascending | Descending |
|---|---|---|
| X polarisation | −0.97 | −0.88 |
| Y polarisation | −0.96 | −0.74 |

This high (negative) correlation factor between bias and the NIR calibration parameter suggests that the NIR variations present in NIR calibration parameter Tna are not real, but artefacts established by some non-ideality in the instrument model, and, further, that the NIR unit reference temperature Tna is extremely stable.

NIR calibration is performed during external manoeuvres, during which the instrument points upwards to the cold sky [5]. During this process, the temperature of the NIR antennas' patches gets colder and outside the nominal temperature range of the instrument. Clearly, the current NIR instrument model, and especially its thermal parametrization, is not able to account for such circumstances. This realisation introduced two main changes to the SMOS calibration. On one hand, starting in 2014, SMOS NIR calibration manoeuvre has been done keeping the Sun at approximately 10 degrees above the antenna plane to avoid getting in a thermal range different from the one during science measurements. On the other hand, the NIR parameters Tna and Tnr were set to a fixed value for the third mission reprocessing. These changes have improved the stability of the measurements.

### 2.1.2. Antenna Losses

In SMOS, the NIR antenna losses are divided between the antenna patch ($L_1$) and the feeding circuits in the innermost part of the antenna ($L_2$) as shown in Figure 3. They are at different physical temperatures. The innermost part of the antenna ($T_{p6}$) is within the thermal control, whereas the antenna patch is more exposed to the temperature fluctuations of outer space ($T_{p7}$).

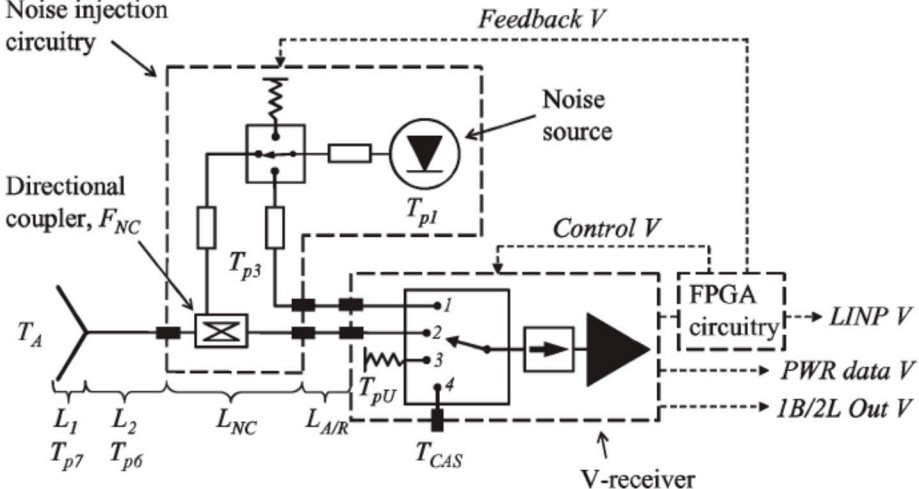

**Figure 3.** Schematic diagram of the structure of the V-channel of the NIR. L stands for loss, Tp for physical temperature, TA for antenna temperature and Tcas for the Calibration Subsystem (CAS) noise temperature [6].

NIR antenna patch losses have been the most challenging problem in SMOS calibration as both $L_1$ and $L_2$ are outside the radiometer's internal calibration loop. A wrong characterisation of the antenna losses introduces variations in the measurements. These variations are related to the variations in the physical temperature of the antenna.

The basic equation for the antenna temperature retrieval at NIR is

$$T_A = -L_1 L_2 T_{NA}\eta + L_1 L_2 [L_{NC} L_A L_{DA}(T_U - T_{t2}) - T_{t1}],\qquad(1)$$

where $T_A$ is the antenna temperature as measured by the NIR, $L_1$ is the antenna patch loss, $L_2$ is the intermediate layer antenna loss, $\eta$ is the NIR pulse length, $L_{nc}$, $L_A$ and $L_{DA}$ are losses of different sections of the cables connecting the antenna to the receiver, $T_U$ is the load noise temperature and $T_{t1}$ and $T_{t2}$ are

$$T_{t1} = \frac{L_1 - 1}{L_1 L_2}T_{p7} + \frac{L_2 - 1}{L_2}T_{p6},\qquad(2)$$

$$T_{t2} = \frac{(L_{NC} - 1)}{L_{NC} L_A L_{DA}}T_{p3} + \frac{(L_A - 1)}{L_A L_{DA}}T_{Cab} + \frac{(L_{DA} - 1)}{L_{DA}}T_{pU},\qquad(3)$$

and $T_{NA}$ is the value measured during calibration, and corresponds to

$$T_{NA} = \frac{-T_{A,cal} + L_1 L_2 \big[L_{NC} L_A L_{DA}\big(T_{U,cal} - T_{t2,cal}\big) - T_{t1,cal}\big]}{\eta L_1 L_2},\qquad(4)$$

where "*X,cal*" indicates the value of parameter "*X*" obtained during the calibration against the cold sky.

Now, if we analyse the equation as a function of the $L_1$ uncertainty using error propagation, we get

$$\frac{\Delta T_A}{\Delta L_1} = \frac{\partial T_A}{\partial L_1} + \frac{\partial T_A}{\partial T_{t1}}\frac{\partial T_{t1}}{\partial L_1} + \frac{\partial T_A}{\partial T_{NA}}\frac{\partial T_{NA}}{\partial L_1},\qquad(5)$$

and finally

$$\Delta T_A = \frac{\Delta L_1}{L_1}\left[\frac{L_1 L_2[TL_D - T_{t1}] - T_A}{L_1 L_2[TL_{D,cal} - T_{t1,cal}] - T_{A,cal}}\left(T_{p7,cal} - T_{A,cal}\right) - \left(T_{p7} - T_A\right)\right], \tag{6}$$

where

$$TL_D = L_{NC}L_A L_{DA}(T_U - T_{t2}), \tag{7}$$

This equation, as given, is difficult to interpret, but by making some realistic numerical simulations, we realised that in a scenario where the calibration was obtained at a $T_{p7}$ of 295 K, errors in the $L_1$ antenna patch loss would propagate to $T_A$ at a different rate depending on the $T_{p7}$ during measurement. Figure 4 shows how an error in the antenna losses characterisation will introduce an error in the antenna temperature that will be a function of the temperature of the antenna patch ($T_{p7}$).

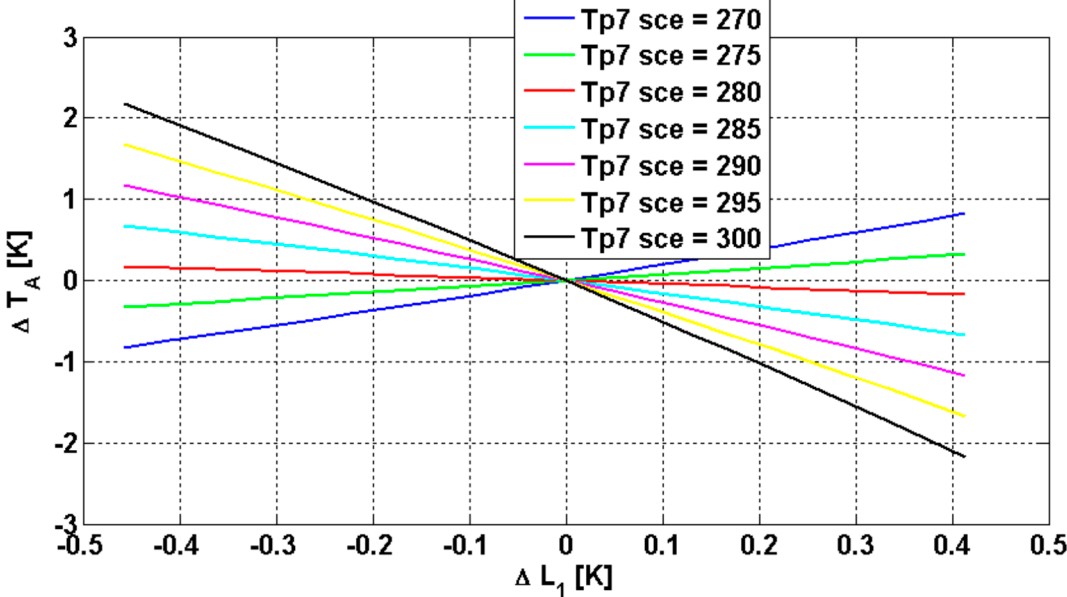

**Figure 4.** Expected error in the antenna temperature as a function of the error in the antenna losses ($L_1$), for simulated scenarios with different temperatures, when the calibration of the NIR was obtained at a temperature of $T_{p7}$ = 295 K.

Therefore, errors in the antenna loss characterisation should be correlated with the variations of the antenna physical temperature, which is exactly what has been observed in SMOS.

Initially, just after launch, the on-ground characterisation values for $L_1$ and $L_2$ were used. Later, during SMOS's first mission reprocessing, an antenna thermal model was introduced to correct for variations observed during NIR external calibration manoeuvres. However, the antenna thermal model was quickly abandoned, as the instrument became more stable following the initial months in orbit. For the second mission reprocessing, the team derived a method to calibrate the antenna losses in orbit [8]. Antenna losses were measured every 15 days, and, since the values were stable, the average value was used for the entire reprocessing. This correction was key to improve the stability of the data in the second mission reprocessing. The calibration procedure could only measure the antenna losses for whole of the antenna patch and the inner part of the antenna ($L_1$ and $L_2$ losses respectively). But the antenna patch and the innermost part of the antenna in SMOS suffer different temperature excursions. Introducing the correct split in the total antenna loss between $L_1$ and $L_2$ is key for obtaining good instrument stability. This split was obtained by assessing the brightness temperature variations over the ocean against an ocean forward model for Stokes-1 measurements and applying the same antenna loss value at the H and V polarisations.

In the third mission reprocessing, it became evident that a different split was necessary for H and V polarisation, as the antenna has different patch for each polarisation. The exercise was then repeated for each of the two polarisations [9]. Figure 5 shows the variations in the brightness temperature biases over ocean as a function of the physical temperature differences between the antenna patch and the innermost part of the antenna, when the $L_1$ antenna loss has been set to 0 dB. The plots show a clear slope in the data, which can account for the antenna losses. NIR-CA H pol $L_1$ antenna loss was set to 0.27 dB, and V polarisation $L_1$ was set to 0.14 dB. $L_2$ values were set to the difference between the total loss as measured by calibration and the corresponding $L_1$ values ($L_2$ equals 0.19 for H and 0.30 for V polarisation for NIR-CA; the other two NIR units are not used to derive the antenna temperature, but their values can be found in [9]).

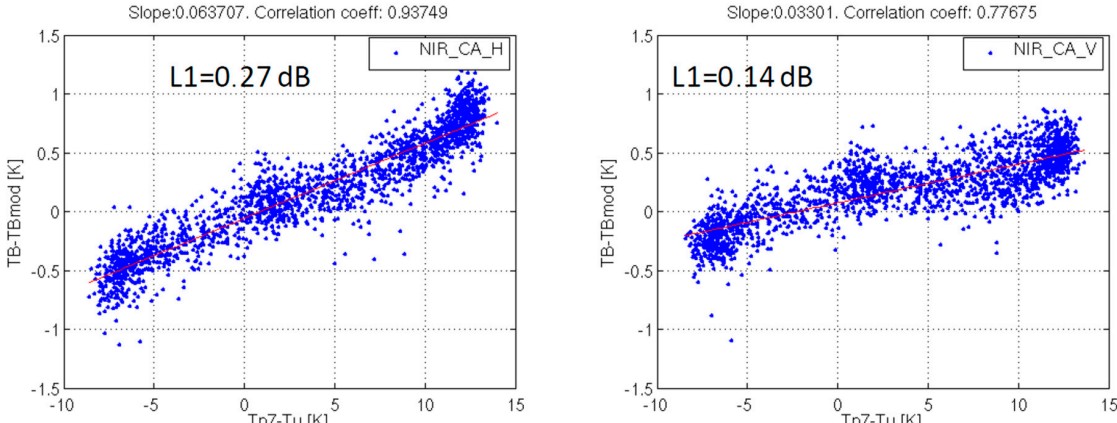

**Figure 5.** Variations of brightness temperature biases over the ocean as a function of temperature variations between the antenna patch and the innermost part of the antenna, for H polarisation (**left**) and V polarisation (**right**).

### 2.1.3. PMS Sensitivities

The sensitivity of the power measurement system (PMS) gain to physical temperature variations was first characterised by the receiver supplier and later verified on-ground at instrument level during the test in thermal vacuum conditions at the Large Space Simulator (LSS) at ESTEC [10], and then again during special calibration events in the SMOS commissioning phase [11]. The latter values have been used until now for adjusting the temperature sensitivity. However, a recent analysis of the variations of the PMS gain through the years showed that the pre-launch PMS sensitivities provide for a more natural behaviour of the PMS gain's aging with time. Figure 6 shows PMS behaviour for receiver LICEF C10 (LICEF stands for lightweight cost-effective front end). As it is seen, using the pre-launch sensitivities provides the best cancellation of the PMS gain oscillations due to physical temperature swings. Similar results are seen in other receivers. For the third mission reprocessing, PMS sensitivities' pre-launch values were used again.

### 2.1.4. PMS Heater Correction

A known problem in the SMOS instrument, which was detected during the thermal tests at the LSS chamber, was PMS offsets jumps following the instrument heater switching from on to off and vice versa. A correction was introduced early in the mission to mitigate this effect, which consisted of a delayed voltage offset with respect to the heater status transition, but the problems were still noticeable, particularly for a few receivers [11].

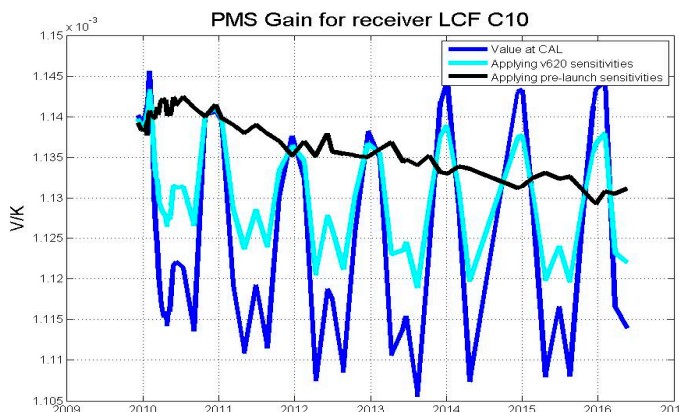

**Figure 6.** SMOS PMS gain for receiver LCF_C10 over the years. The blue line indicates the value as measured during the calibration event at the physical temperature during the calibration event. The cyan line and black line show the PMS gain transported to 21 degrees Celsius using the second mission reprocessing and pre-launch PMS sensitivities, respectively.

A more careful analysis showed that the jumps related to the heater status do not correspond to a simple delayed offset, but that the behaviour follows a double exponential [9]. Figure 7 shows the PMS voltages for a calibration event, where a constant noise from an internal warm load source was introduced at the receivers.

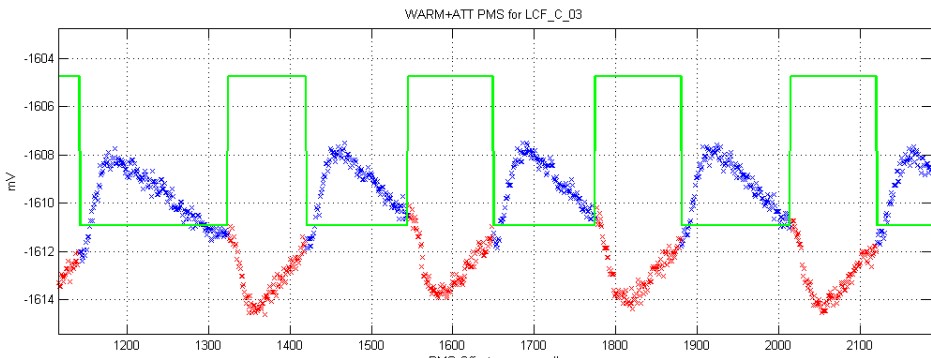

**Figure 7.** PMS voltages for LICEF C-03 when a constant noise source is introduced. In green, the status of the heater is depicted. Red crosses for the PMS voltage indicate that the heater is on and blue crosses show when the heater is off.

Based on this analysis, the correction applied, $\Delta V$, was set to

$$\Delta V_{ON} = \alpha_{ONi}(V_i - V_{max})\left(1 - e^{\frac{-t}{\tau_{ON\_1i}}}\right) + \beta_{ONi}(V_i - V_{max})\left(1 - e^{\frac{-t}{\tau_{ON\_2i}}}\right), \tag{8}$$

$$\Delta V_{OFF} = \alpha_{OFFi}(V_i - V_{max})e^{\frac{-t}{\tau_{OFF\_1i}}} + \beta_{OFFi}(V_i - V_{max})e^{\frac{-t}{\tau_{OFF\_2i}}}, \tag{9}$$

where $V_i$ is the current PMS value without correction for each of the 72 receivers, in volts. $V_{max}$ is the maximum measurable PMS, set to a value of 2.5V. $\alpha$, $\beta$, $\tau_{ON}$ and $\tau_{OFF}$ are fixed constants for each receiver that empirically determine the double exponential behaviour, and *t* is the number of epochs since the corresponding transition of the heater status (on to off, or vice versa).

The validation of this correction was performed by means of a relative comparison of the antenna temperature of one receiver to the average of all receivers. Figure 8 shows the behaviour for L1OP v620 (delay heater correction) and for L1OP v724 (double exponential heater correction). The new correction clearly reduces the obvious PMS offset jumps due to the heater status.

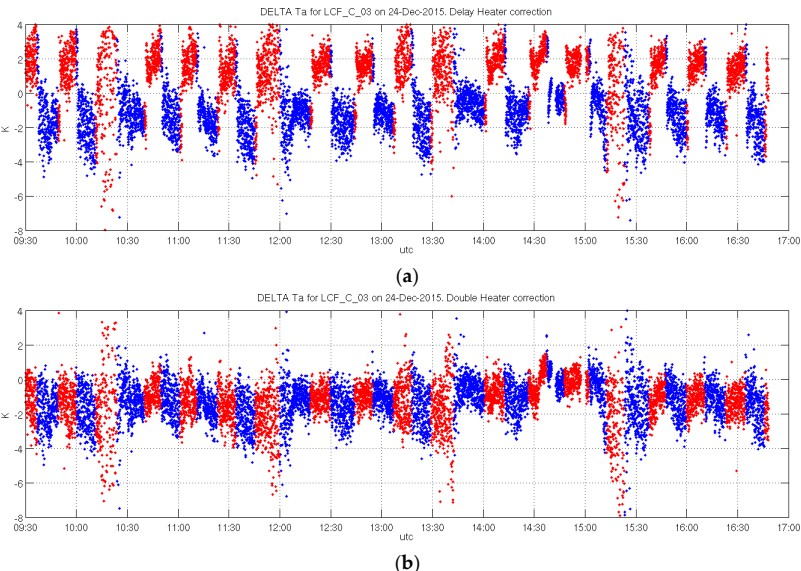

**Figure 8.** Difference between the antenna temperature measurement of receiver LCF_C_03, with respect to the average of all receivers, when applying the v620 delayed offset heater correction (**a**) and when applying the new v724 double exponential heater correction (**b**). The colour of the points in both plots indicates the status of the heater: red when the heater is on and blue when it is off.

#### 2.1.5. Antenna Patch Thermistor Correction

Another aspect that was discovered during the second mission reprocessing analysis was an increased bias immediately following the Sun's eclipse by the Earth, relative to the instrument. This effect clearly pointed to another problem related to temperature variations. While the instrument backend is kept under thermal control [McMullan et al., 2008], the antenna patches suffer large thermal excursions. Those changes are monitored by three thermistors placed at the screw of each NIR antenna patch ($T_{p7}$ in Figure 3) and are used in the NIR radiometric equation presented in Section 2.1.2.

The team considered that the reading of the thermistor did not properly describe the temperature of the antenna patch and proposed a correction [12]. The correction was introduced based on the observed thermal latency during inertial external manoeuvres. During these manoeuvres, the instrument points at the cold sky for several minutes. However, $T_{p7}$ thermistor readings take a long time to stabilise to a constant temperature. The thermistor reading was considered to be thermally coupled to the innermost part of the antenna through the antenna screw, inside whose head the thermistor is mounted. As such, the thermistor reading is not fully representative of the antenna patch region. The team decided then to apply a correction to the thermistor reading by assuming that the temperature to which the thermistor stabilises at the end of the external manoeuvre is the actual temperature during the entire inertial manoeuvre. The following correction was derived:

$$\hat{T_{p7}} = T_{p7} - \frac{1}{LP}\frac{dT_{p7}}{dt},\tag{10}$$

where $\hat{T_{p7}}$ is the corrected thermistor temperature, $T_{p7}$ the actual thermistor reading, and $LP$ a constant that was estimated to be $-0.0031$.

The correction was then used to process the ocean brightness temperature, and the bias with respect to the forward model was re-assessed. The impact of this correction is a clear mitigation of the bias observed during the eclipse period. Figure 9 shows the Y polarisation brightness temperature bias observed over the ocean with and without the $T_{p7}$ correction applied. The increased bias in the eclipse is observed around 35N to 60N degrees in latitude during the Northern Hemisphere (NH) winter months in the left plot.

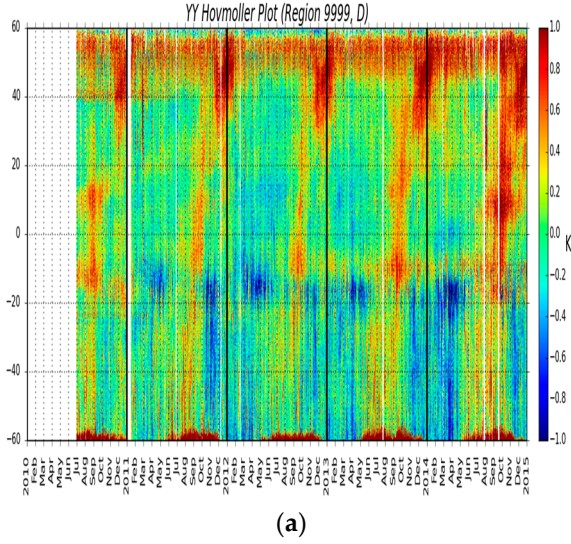

(**a**)

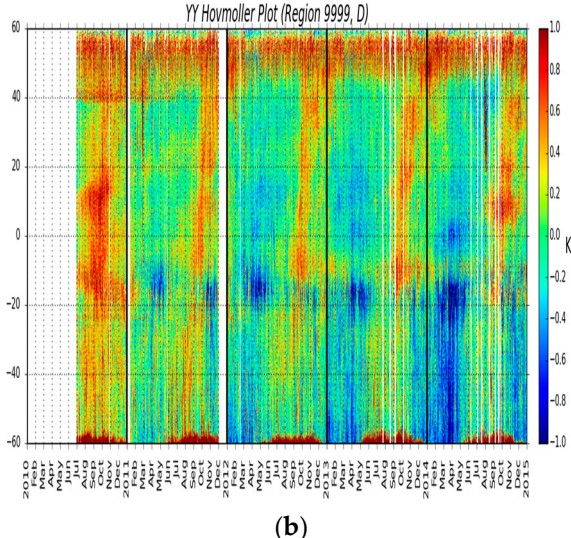

(**b**)

**Figure 9.** Hovmoller plot showing Y pol BT bias over the ocean as a function of time and latitude with (**a**) and without (**b**) the $T_{p7}$ correction.

## 2.2. Changes in Image Reconstruction

Image reconstruction is a process where the calibrated MIRAS data are turned into radiometric maps that can be projected on the Earth's surface. For the v724 processing baseline, three main improvements were made to this process. They are improvements related to

- Gibbs phenomena correction.
- The use of a sea-ice mask.
- Correction of the Sun's influence in the images.

In the following sub-sections, we describe these updates in detail.

### 2.2.1. Gibbs-2 Algorithm

The so-called Gibbs-2 algorithm is an evolution of the Gibbs-1 algorithm applied to SMOS measurements since its launch. Originally, the Gibbs correction aimed to reduce the Gibbs artefacts that appeared in the image following large BT transitions between land and ocean (or sky and Earth) due

to limited coverage in the visibility domain. Soon after, the team realised that the correction not only reduces the Gibbs artefacts but also a floor error induced in the retrieved images due to dissimilarities in the antenna patterns and the aliasing [13,14]. Gibbs-1 correction reduces this so-called floor error by removing a constant brightness temperature (BT) in the reconstruction process, which reduces the visibility values before inversion, and adding it back at the end of the inversion process. In Gibbs-2, the process has evolved to include the use of an artificial scene as close as possible to the observed one. This artificial scene, $V_a$, uses a Fresnel model over the ocean and a constant value (250 K) over land. Figure 10 shows an example of the artificial scene as used in the image reconstruction processor. The visibilities of this artificial scene are computed using an SMOS instrument model:

$$V_a = GT_a, \tag{11}$$

where $T_a$ represents the modelled BT of the artificial scene, $G$ is the instrument model, and $V_a$ are the visibilities derived from the $T_a$ scene.

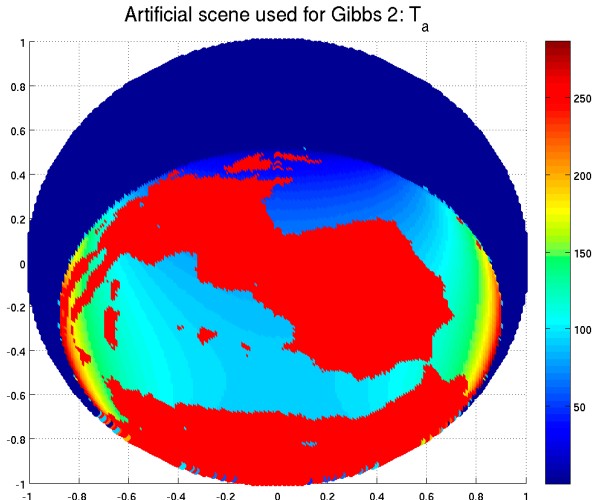

**Figure 10.** Artificial scene of one SMOS observation of the Iberian peninsula and northern Africa.

Then, the image reconstruction algorithm is applied over the following differential linear problem:

$$V - V_a = G(T - T_a), \tag{12}$$

yielding to the following retrieved BT:

$$T_r = U^*ZJ^+(V - V_a) + T_a, \tag{13}$$

where $U$ is the Fourier transform operator, $Z$ is the zero-padding operator beyond the SMOS frequency coverage, $J = GU^*Z$ is the image reconstruction operator used in the SMOS processor and $J^+$ is the pseudo-inverse of $J$ [15].

### 2.2.2. Sea-Ice Mask

The calculation of the artificial scene used in the Gibbs-2 algorithm described above is based on the use of a fixed global land–ocean mask. In fact, we identify the land and ocean pixels within the field of view and assign a constant value over land and the Fresnel forward model over ocean. To improve the accuracy of the artificial scene in seasonal sea-ice growth, we have developed an operational strategy to measure the sea-ice extension from the actual SMOS measurements and apply this extension to the artificial scene. Measurements are collected for 10 days, then a mask of the percentage of sea ice over the ocean is derived and this mask is used in the Gibbs-2 algorithm with a constant value of 250 K

(same as for land pixels), as can be seen in Figure 11. In the third mission reprocessing data, the mask computation will be aligned with the data that is applied. However, in nominal operations, the mask will be applied, typically with a 12-day delay from the moment it first started estimating the extension. Errors derived from this 12-day delay were analysed and resulted to be much lower than those present when not applying the correction at all.

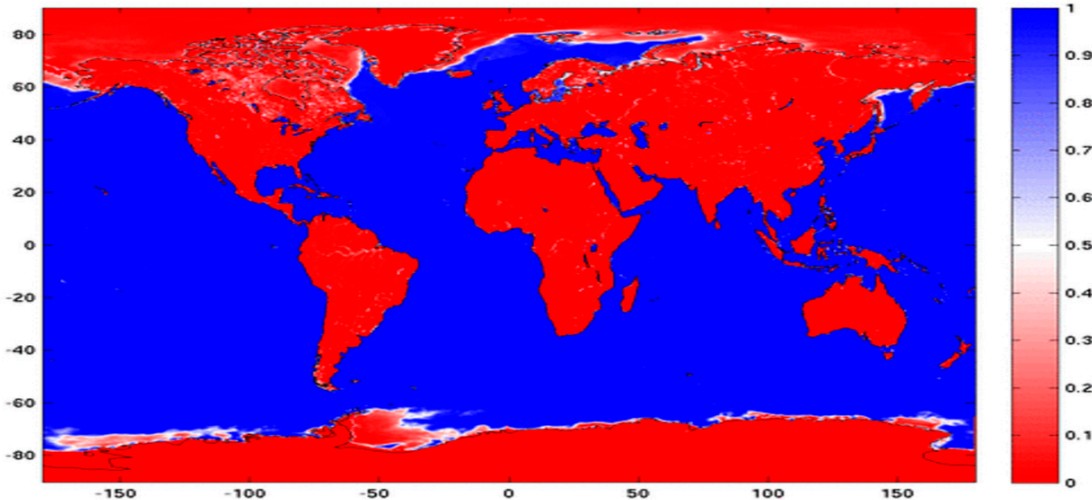

**Figure 11.** Mask showing the extensions of land/ice pixels versus ocean pixels, as derived from January 2010 SMOS BT measurements.

Figure 11 shows that the sea ice detected by SMOS BT measurements goes beyond the continental surfaces.

### 2.2.3. Super-Sampled Sun Correction

L-band observations of the Sun disk showed that its BT emissivity is not spatially homogeneous [16]. Sunspots tend to have much larger BT emissions than other Sun regions. The Sun correction applied to SMOS during the second mission reprocessing considered the Sun as a point source. The team considered this correction to be insufficient and derived a new method to correct for the Sun BT emissions to take into account those spatial inhomogeneities Figure 12.

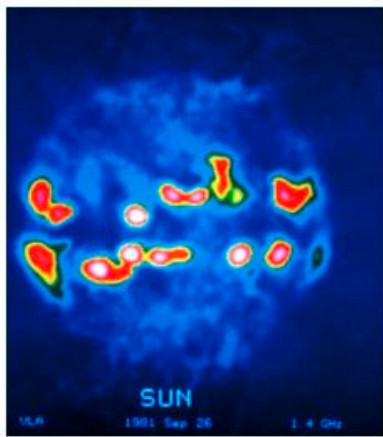

**Figure 12.** L-band BT emission of the Sun [16].

The so-called super-sampled Sun correction applied in L1OP v724 estimates the BT of multiple spots within the Sun disk, minimising the differences between the simulated signal and the BT observations in an area around the Sun and in the Sun tails [17].

$$V_{Sun} = \sum_i V_{1ki} T_i,\tag{14}$$

where $V_{Sun}$ are the visibilities of the Sun as a function of the 1K visibilities ($V_{1ki}$) computed for each of the sub-sample points' times and $T_i$ corresponds to the BT estimated for each of those points from the following equation:

$$
\begin{bmatrix}
F^{-1}(V - V_{G2})_1 - \langle F^{-1}(V - V_{G2})_0 \rangle \\
F^{-1}(V - V_{G2})_j - \langle F^{-1}(V - V_{G2})_0 \rangle \\
F^{-1}(V - V_{G2})_{n^*} - \langle F^{-1}(V - V_{G2})_0 \rangle \\
\dots \\
\frac{T_0}{wnPOS} \\
\dots \\
\frac{T_0}{w}
\end{bmatrix}
=
\begin{bmatrix}
F^{-1}(V_{1K1})_1 & F^{-1}(V_{1Ki})_1 & F^{-1}(V_{1KnPOS})_1 \\
F^{-1}(V_{1K1})_{n^*} & F^{-1}(V_{1Ki})_{n^*} & F^{-1}(V_{1KnPOS})_{n^*} \\
1/w & \dots & 0 \\
\dots & 1/w & \dots \\
0 & 0 & 1/w \\
1/w & \dots & 1/w
\end{bmatrix}
[T_i]\tag{15}
$$

where:

- $F^{-1}(V - V_{G2})_1$ is the inverse Fourier transform of the difference between the measured visibilities and the Gibbs-2 synthetic visibilities;
- $\langle F^{-1}(V - V_{G2})_0 \rangle$ represents the average value over the clean 'o' pixels surrounding the sun disc and is calculated as the average of the inverse Fourier transform of the difference between the measured visibilities and the Gibbs-2 synthetic visibilities.
- $T_0$ is the first estimate of the Sun temperature, obtained assuming the Sun as a point source.
- w is a weight to be finely tuned to get the best compromise in condition number versus sensitivity (for now fixed at $10^6$).
- nPOS is the number of over-sampled points, fixed to 37 in the L1OP.
- $1,..,j,..,n$ is the number of points polluted by the solar radiation, including the disc of the sun and the tails.
- $V_{1K1}, \dots, V_{1KnPOS}$ are the system response functions calculated over the over-sampled grid.
- $F^{-1}(V_{1Ki})_{n^*}$ is the inverse Fourier transform of $V_{1K1}$ calculated over the polluted pixel.
- $T_i$ is the set of estimated temperatures for the 37 points of the sun disc. It has been shown in [17] that the solution to this problem is explicit.

This correction succeeds in better reducing the residuals of solar radiations, as shown in Figure 13.

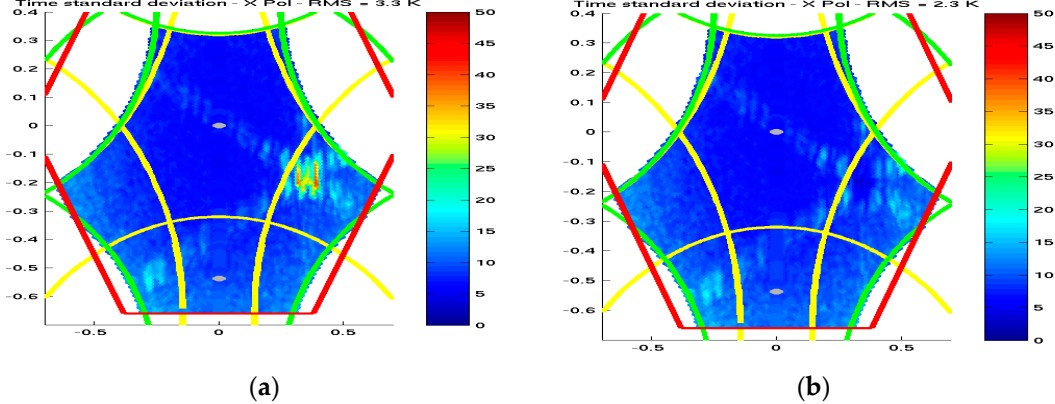

**Figure 13.** Time standard deviation of 100 snapshots over the ocean, with the Sun alias present at approximately [0.4, −0.15], with the old Sun correction (**a**) and the super-sampled correction (**b**).

Figure 13 shows that the new super-sampled Sun correction reduces the variations in the measurements along the Sun tails and within the Sun alias disk.

### 2.2.4. Sun Correction in the Back

SMOS measurements showed that the radiation coming from the Sun is observed even in the case the Sun is behind the antenna plane, through the antenna back-lobes [18].

The team considered that it was important to correct for this foreign source radiation and applied the Sun correction algorithm described in [19] and later modified in [20], even in the case the Sun was behind the antenna. Figure 14 shows the bias observed in the data before and after the extension of the Sun correction in the back.

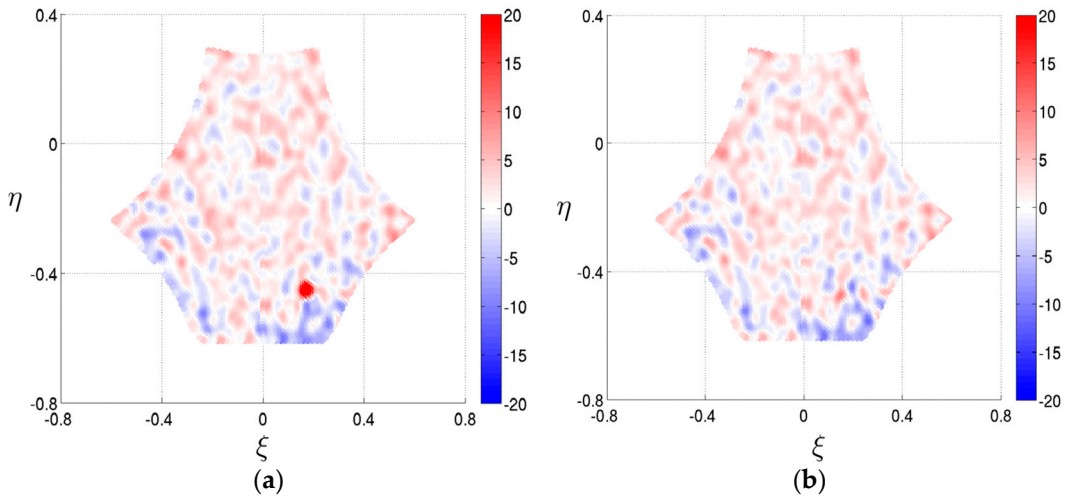

**Figure 14.** SMOS biases from a measurement in the Pacific after removing the expected forward model when the Sun is in the back of the instrument before (**a**) and after (**b**) the Sun BT correction in the back [18].

### 3. Results and Discussion

The changes applied in the v724 version of the processor have been analysed using a dedicated end-to-end processing campaign that involved 3 years of data. The results presented hereafter show the improvement of the data quality in different metrics, such as reduction of the spatial biases, improvement of stability, reduction of land–sea contamination biases for certain polarisations, better match to in-situ measurements and reduction of the $\chi^2$ in the soil moisture retrievals.

### 3.1. Orbital and Seasonal Stability

The quality of the stability of the measurements is established by comparison with the ocean forward model. In this case, one of the metrics used by the SMOS team is the one provided by the Hovmoller plots showing the biases observed in the Pacific open ocean in time and latitude. This metric allows us to assess both the orbital stability (variation along the vertical axis) and the seasonal stability (variation along the horizontal axis). The analysis is done independently per polarisation and separately for ascending and descending passes.

Figure 15 shows an example of the stability of the measurements in the second mission reprocessing [21] and the expected behaviour for the third mission reprocessing, for both X and Y polarisations. Descending orbits suffer the most from larger instrument thermal dynamics and are always more prone to measurement instabilities.

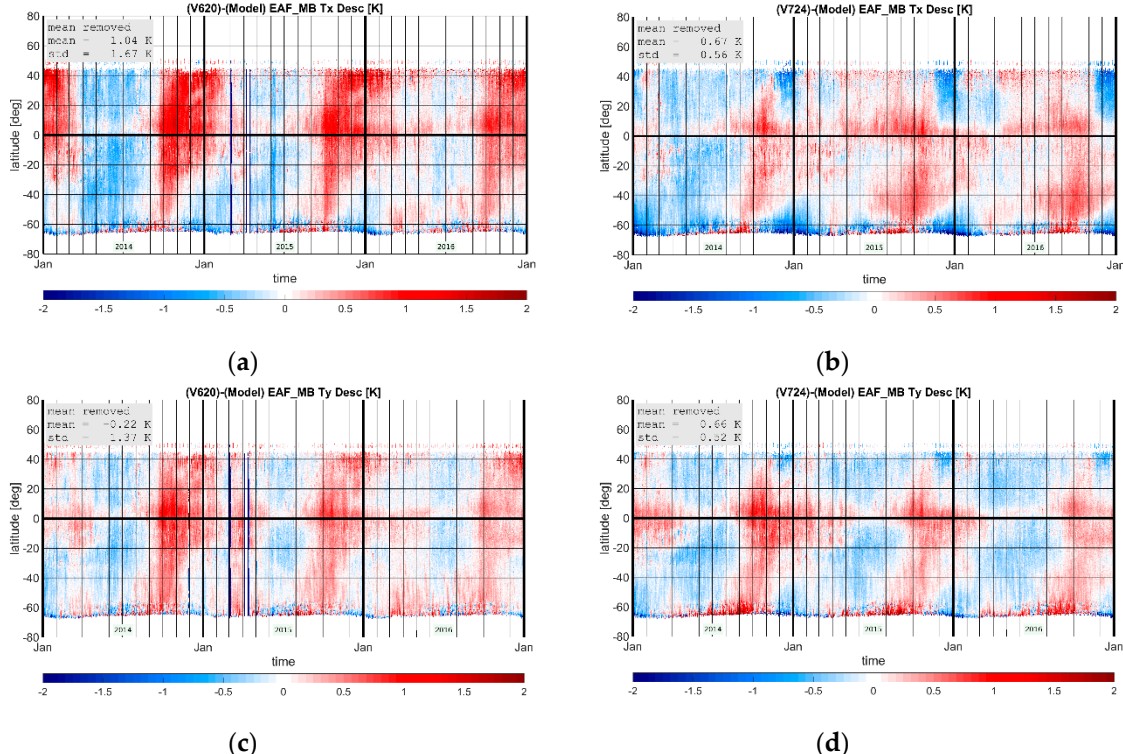

**Figure 15.** Hovmoller plots showing the bias between measurement and model averaged over the entire extended alias-free field of view for descending orbits for (**a**) v620 X polarization, (**b**) v724 X polarization, (**c**) v620 Y polarization and (**d**) v724 Y polarization.

Figure 15 shows that both the orbital and seasonal instabilities have been reduced in the third mission reprocessing. A further metric to assess the improvement in stability is the standard deviation of these Hovmoller plots. The values in Table 2 show an important reduction of the instabilities.

**Table 2.** Mean bias for the v620 data (second mission reprocessing) and the 724 data (third mission reprocessing), computed in the Pacific region between latitudes 45S and 10N.

| Orbit Pass | Polarization | Second Mission Reprocessing | Third Mission Reprocessing |
|---|---|---|---|
| Ascending | X | 1.73 K | 0.57 K |
| | Y | 1.41 K | 0.35 K |
| Descending | X | 1.67 K | 0.56 K |
| | Y | 1.37 K | 0.52 K |

### 3.2. Spatial Biases

Spatial biases have been one of the largest challenges in the SMOS image reconstruction process [21, 22]. The Level 2 Ocean Salinity Processor uses the ocean target transformation (OTT) technique to reduce them [23,24], but this technique only works well in scenes whose brightness temperature is roughly stable, such as measurements in the open ocean. Near the coastlines, or for any measurement over land, the technique does not work properly. Therefore, it is of utmost importance that the spatial biases are minimised at the image reconstruction level. The changes introduced in the v724 processor have considerably reduced the spatial biases in the extended alias-free field of view.

Figure 16 shows the spatial biases for X and Y polarisation. A very important aspect to note in the spatial bias improvement in Y polarisation is the reduction of a negative gradient from top to bottom of the OTT image.

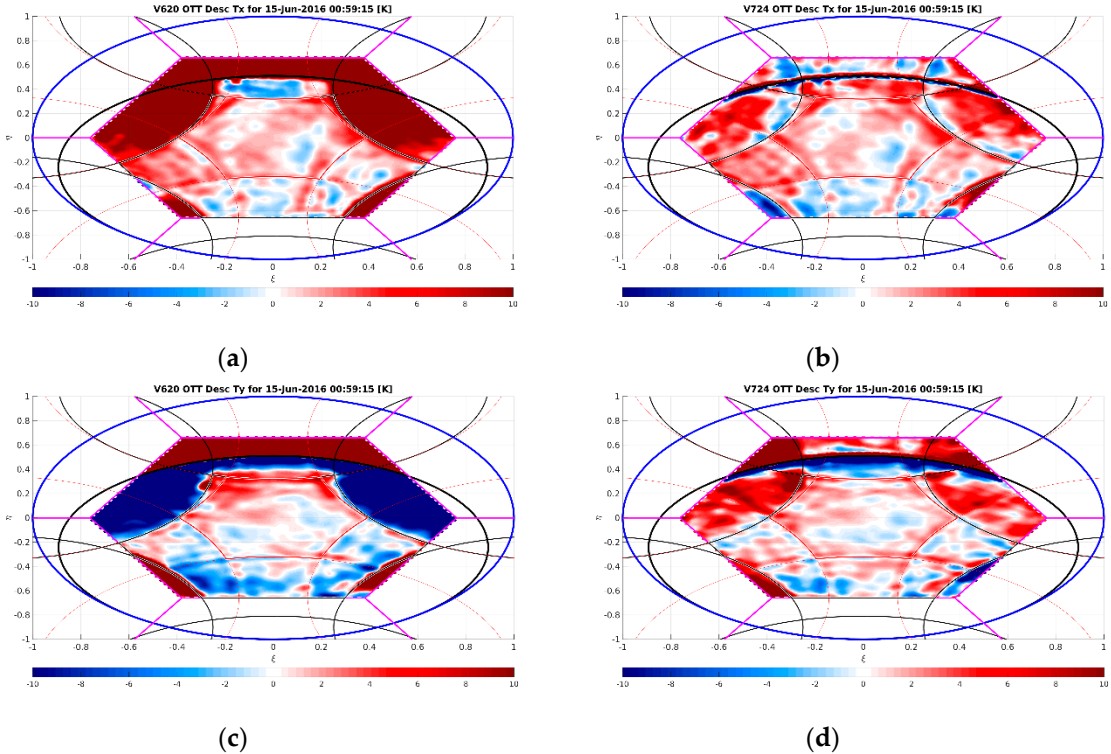

**Figure 16.** Spatial biases for descending orbits in (**a**) v620 X polarization, (**b**) v724 X polarization, (**c**) v620 Y polarization and (**d**) v724 Y polarization.

### 3.3. Land–Sea Contamination

Another critical aspect of the SMOS radiometric measurements is the land–sea contamination. This refers to the increase in the bias that occurs in the ocean measurements near land masses and vice versa. The adjustments in calibration and the new Gibbs-2 image reconstruction technique have achieved a significant improvement in land–sea contamination, even though this is still substantially present in the third mission reprocessing. Figure 17 shows the biases, over ocean, in the global maps for one particular month of data (June 2016) for the four polarisations for the second [21] and third mission reprocessing. The improvements are most noticeable in Y polarisation and in the fourth Stokes parameter. On the other hand, the contamination in Tx has changed but remains at similar levels, and similarly for the third Stokes parameter.

In must be noted that the Level 2 Ocean Salinity Processor includes an empirical correction of the land–sea contamination. Being able to reduce the original bias is an important aspect of the $L_1$ processor, but almost more important is that the residual bias remains constant, which can then be corrected empirically at Level 2. For this reason, another metric assesses the variation of the land–sea contamination bias at Level 1 by means of the standard deviation. Figure 17 shows this metric, over ocean, for Y polarisations only. Similarly to Figure 17, the land–sea contamination variation is substantially reduced, mainly for Y polarisation (Figure 18) and for the fourth Stokes parameter.

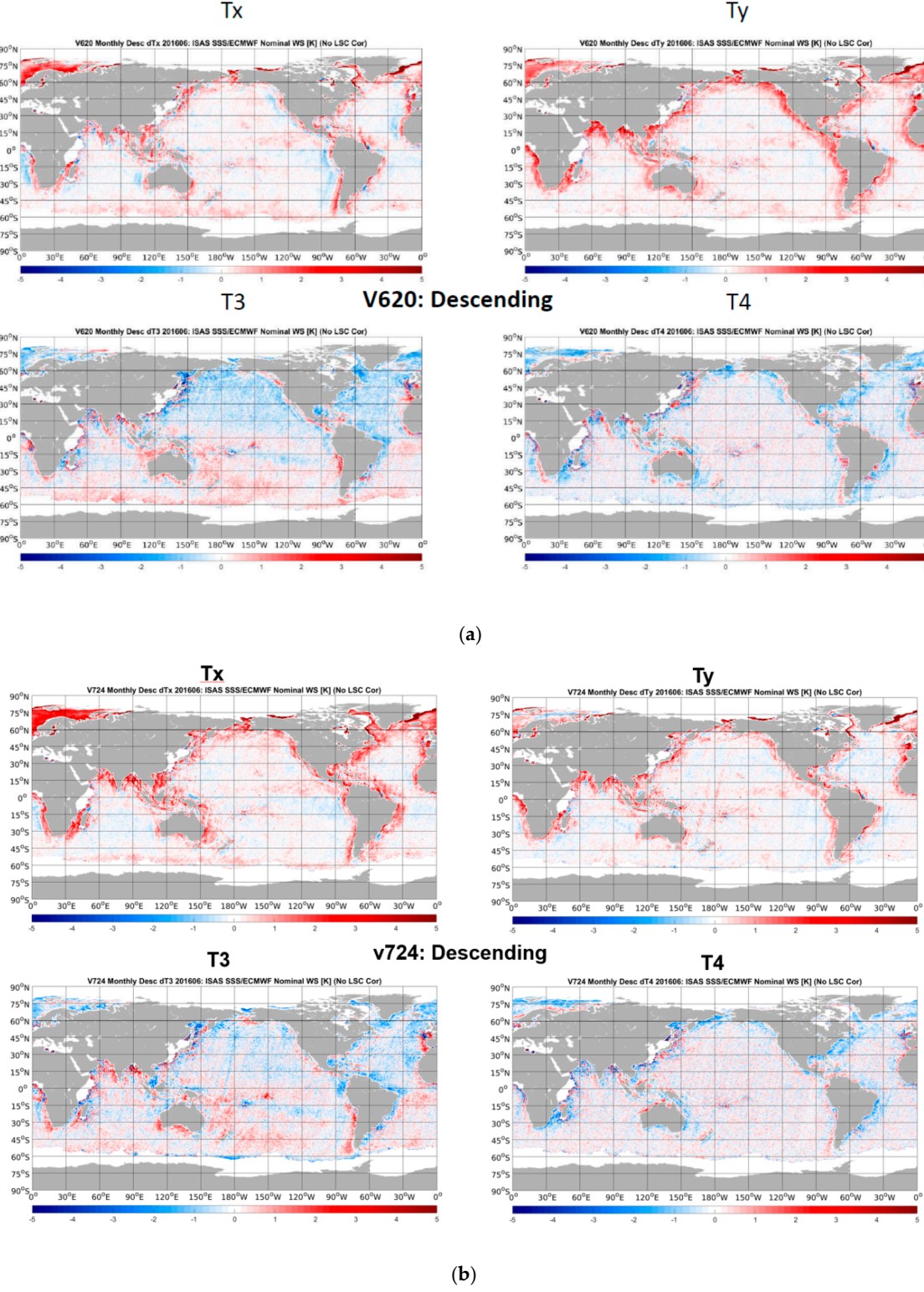

**Figure 17.** Maps of bias between SMOS measurements and the ocean forward model, showing an increase excess of bias in regions near the coast, known as land–sea contamination. (**a**) all four polarizations for v620 and (**b**) all four polarisations in v724.

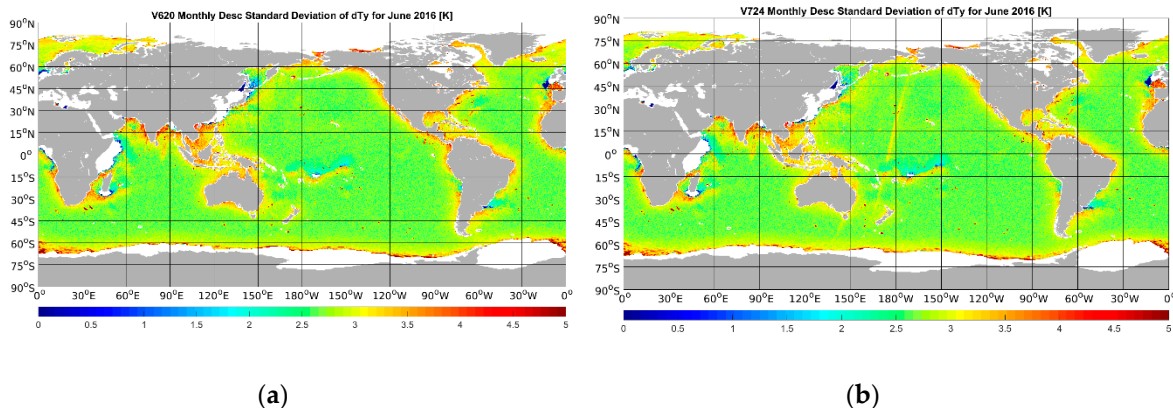

**Figure 18.** Standard deviation of the SMOS measurements in June 2016 for v620 (**a**) and v724 (**b**).

### 3.4. Impact on Retrieved Soil Moisture and Vegetation Optical Depth (VOD)

As part of our standard metric protocol, new versions of $L_1$ data are systematically processed with the Level 2 soil moisture processor to assess the changes compared to the previous $L_1$ processor version. This assessment is made through spatial monthly maps showing the changes on retrieved soil moisture and retrieved opacity along with $\chi^2$ changes. A second perspective is obtained through time series of retrieved soil moisture corresponding to a collection of in-situ time series of measured soil moisture for the two-year period 2011–2012 and provides quantitative metrics but for a limited number of grid points.

For the purposes of this analysis, the same Level 2 Soil Moisture v650 processor has been used in order to assess only the improvements in the $L_1$ processor.

#### 3.4.1. Spatial Maps of Retrieved Soil Moisture and Opacity

Figure 19 displays the differences in soil moisture and opacity of v724 minus v620 for the month of June 2014, separated by ascending and descending orbit passes. The overall global change is rather neutral, with mean differences close to 0 but with a significant variability that appears very structured spatially. The significant changes correspond to specific areas, with contrasts between transition areas and forest in both retrieved soil moisture and opacity. Below dense forest v724, soil moisture and opacity tend to decrease, with patterns changing in position between ascending and descending orbits, e.g., the North American east coast, Amazonian forest, and African Congo forest. This is probably a signature of the Gibbs-2 correction, as the contrast of land/sea masses is not similar within the SMOS field of view for these locations depending on the orbit pass.

The L1C v724 data generate significant changes compared to the L1C v720, and the question whether those changes are in the right direction is addressed by the two following sections.

#### 3.4.2. $\chi^2$ Test

$\chi^2$ is an important metric to assess the quality of the soil moisture retrievals and is used widely in many retrieval processes. It provides a measure of the agreement (best fit) between the geophysical modelling that resulted in the retrieved parameters and the $L_1$ data that were used accounting for the expected noise on the observed data. In this study, we considered rather the reduced $\chi_r^2$ form, which is $\chi^2$ divided by the number of degrees of freedom. Using $\chi_r^2$ introduces a normalisation, which is preferable as the Level 2 processor includes $L_1$ data filtering that may result in slightly different numbers of degrees of freedom between the two $L_1$ datasets.

Figure 20 shows the changes of the $\chi_r^2$ between v620 and v724, computed as the ratio $\chi_r^2$ v724 divided by $\chi_r^2$ v620 for June 2014 for ascending and descending orbits. Very similar maps are observed at other months of the year. Ratios $\gg 1$ (toward red colours) indicate degraded (increased) v724 $\chi_r^2$

with respect to v620 and ratios << 1 (toward blue colours) indicate improved (decreased) v724 $\chi_r^2$ with respect to v620.

The team analysed the changes in $\chi_r^2$ from v620 to v724 over land and concluded that $\chi_r^2$ has improved (reduced) significantly over most of the globe. Most exceptions are either neutral (light blueish/reddish area) or are related to presence of radio frequency interference (RFI), especially in the Middle East region and South Asia. All maps report a significant improvement (decrease) in v724 $\chi_r^2$ compared to v620 at global scale with distribution ratio modes marker ‣ always below 1 and strong negative asymmetry. Many continental areas show a deep blue colour, which indicates very significant improvements that also appear to be very stable in time for different seasons and different years. Similar to Figure 19, Figure 20 patterns show some differences between ascending and descending orbits. It is important to notice the good match of these blue spatial patterns in Figure 20 with the most significant change patterns in retrieved soil moisture and opacity reported in Figure 19; where v724 introduced the strongest changes in retrieved parameters is also where the best fit has improved the most with reduced $\chi_r^2$. Finally, using the v724 data increases the number of successful retrievals by 2% to 3%.

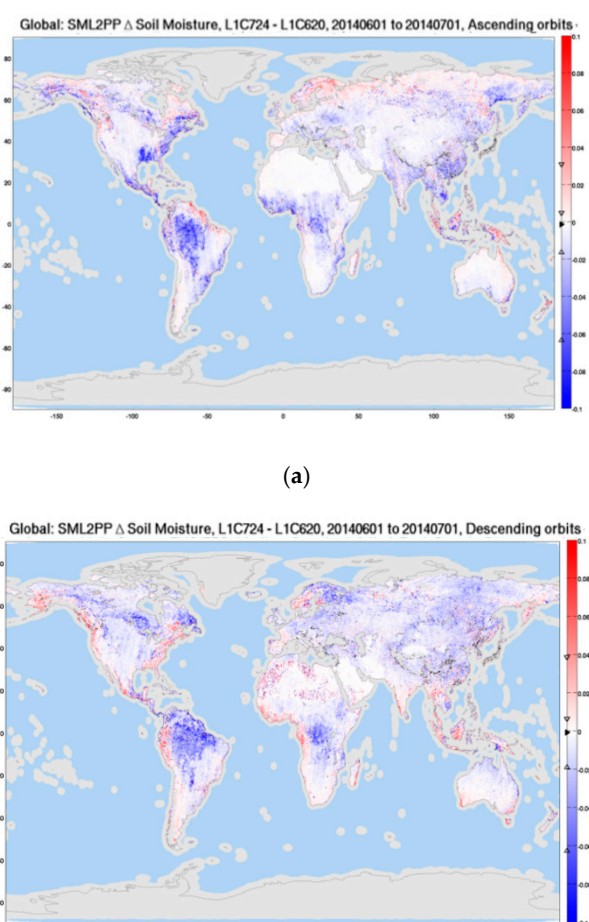

(a)

(b)

**Figure 19.** *Cont.*

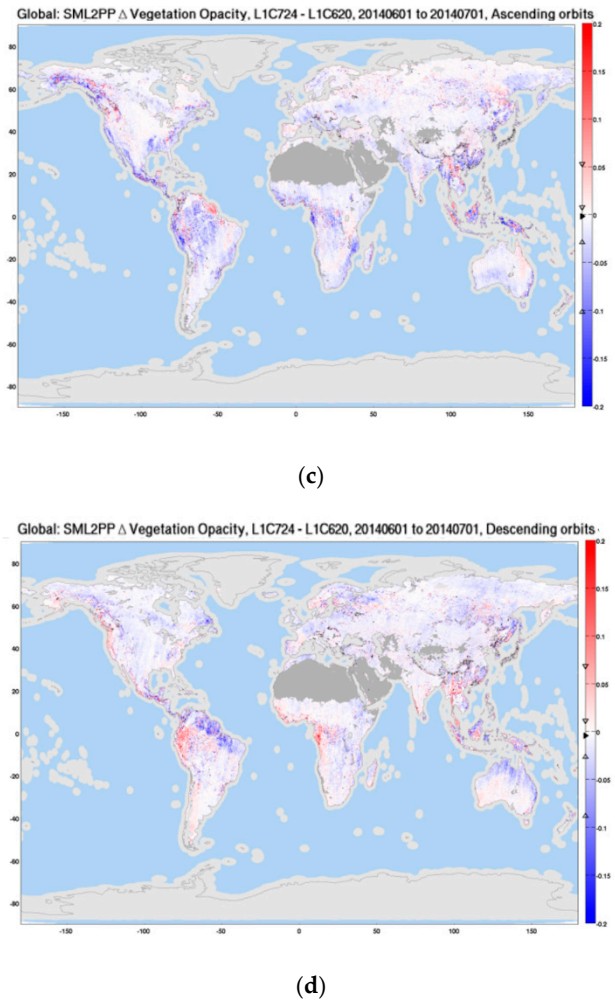

(**c**)

(**d**)

**Figure 19.** Maps of averaged difference, v724–v620, over the month of June 2014 in (**a**) soil moisture for ascending orbits, (**b**) soil moisture in descending orbits, (**c**) VOD in ascending orbits and (**d**) VOD in descending orbitsBlue (resp., red) indicates a decrease (resp., an increase) in V724 soil moisture, VOD compared to v620.

### 3.4.3. In-Situ Soil Moisture

Several networks of in-situ soil moisture measurements stations (ISMN) can be used to assess the quality of SMOS-retrieved soil moisture. SMOS coarse resolution observations and ultra-local in-situ measurement are not necessarily fully comparable, but become useful when assessing relative differences between two versions of processing of the same satellite data. SMOS soil moisture retrievals obtained from L1OP v620 data and from L1OP v724 data are compared against in-situ soil moisture time series of 250 validation sites taken from 11 in-situ soil moisture networks (Figure 21).

SMOS retrieval data and in-situ data are first co-located in space and time by taking the SMOS grid-points closest to the stations and by pairing in time SMOS and in-situ data of less than 7.5 min to a maximum of 30 min of absolute time difference, depending on the in-situ network temporal sampling characteristics. We denote the SMOS and in-situ collocated time series $(S_t, I_t)$ and the associated difference time series $(\Delta_t = S_t - I_t)$.

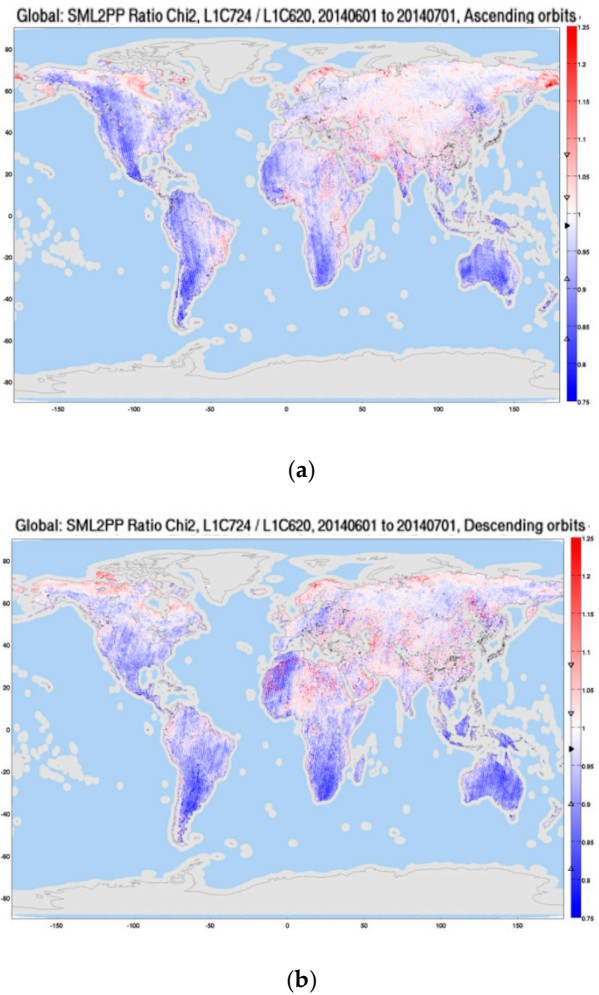

**Figure 20.** Maps of averaged $\chi_r^2$ ratios v724/V620 for the month of June 2014 for ascending orbit (**a**) and descending orbit (**b**). Blue indicates that v724 improves with lower $\chi^2$ compared to v620′. The markers located on the colour bar show the distribution variables, the mode is represented by the ▸ marker. The 68.3% and 95.4% percentiles are shown by the inner and outer ▴ ▾ markers, respectively.

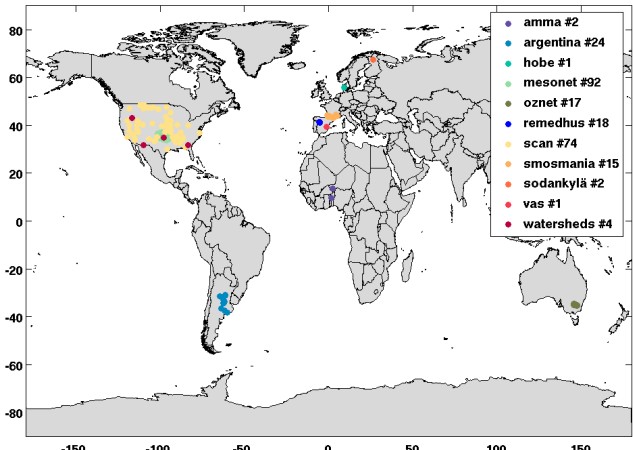

**Figure 21.** The 11 in-situ soil moisture networks and the position of their sites. The legend reports the network name, its associated colour and the number of the sites we considered for a total of 250 sites.

We computed the usual statistics and their 95% confidence intervals (CI95) obtained by bootstrap x to assess the two processor versions. For $(S_t, I_t)$, we computed their means and standard deviations, $\mu_S$ and $\sigma_S$ for SMOS and $\mu_I$ and $\sigma_I$ for in-situ data and the correlation R. For the $\Delta_t$ differences, we computed the bias, the standard deviation (STDD) and the root mean square (RMSD).

The results vary from site to site, but most of them show a better correspondence between the in-situ measurements and the data than the v724 processor or similar performance to the v620 processor. This is reflected by the overall performances, which are obtained by computing the statistics on the concatenation of all sites' time series $(S_t, I_t)$, which are reported in Tables 3 and 4 along with their graphic representation using Taylor diagrams (Figure 22).

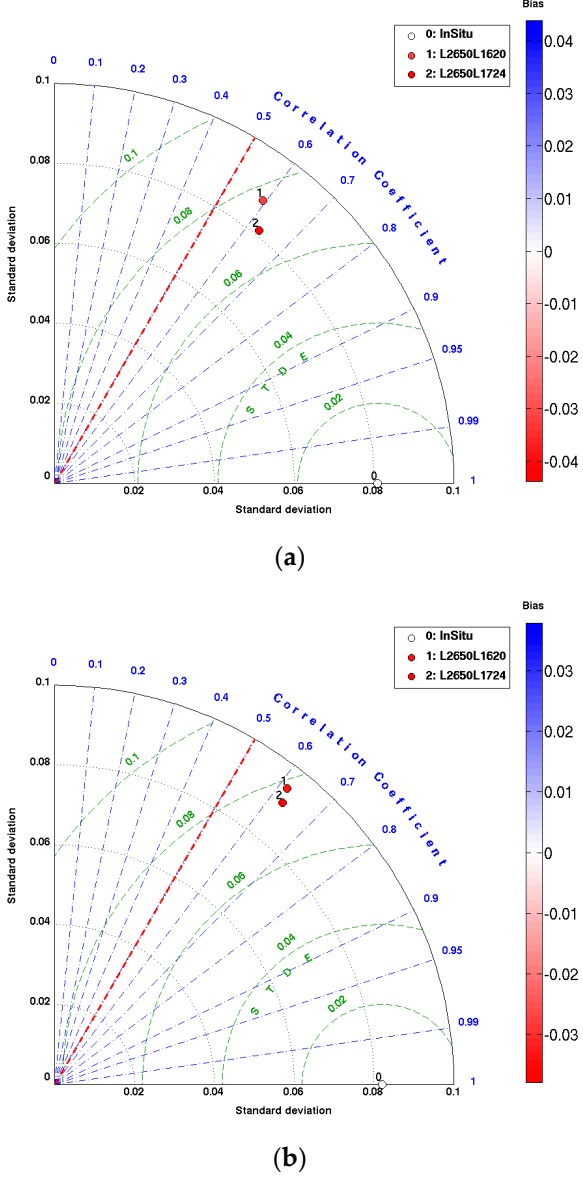

**Figure 22.** Taylor diagram of overall retrieved soil moisture time series with respect to in-situ measurement time series for ascending orbits (**a**) and descending orbits (**b**).

**Table 3.** Δ time series statistics and their CI95, 95% confidence intervals, given between parentheses.

| L1OP | Ascending Orbits | | | | | Descending Orbits | | | | |
|---|---|---|---|---|---|---|---|---|---|---|
| | R | bias | STDD | RMSD | #data | R | bias | STDD | RMSD | #data |
| v620 | 0.59 (0.016) | −0.031 (0.004) | 0.076 (0.004) | 0.083 (0.004) | 38943 | 0.62 (0.017) | −0.034 (0.004) | 0.078 (0.005) | 0.085 (0.004) | 42963 |
| v724 | 0.063 (0.014) | −0.044 (0.004) | 0.070 (0.004) | 0.082 (0.004) | 40108 | 0.63 (0.014) | −0.038 (0.004) | 0.075 (0.004) | 0.084 (0.004) | 44099 |

**Table 4.** SMOS and in-situ time series statistics and their CI95, 95% confidence intervals, given between parentheses.

| L1OP | Ascending Orbits | | | | Descending Orbits | | | |
|---|---|---|---|---|---|---|---|---|
| | $\mu_S$ | $\mu_I$ | $\sigma_S$ | $\sigma_I$ | $\mu_S$ | $\mu_I$ | $\sigma_S$ | $\sigma_I$ |
| v620 | 0.167 (0.004) | 0.198 (0.004) | 0.088 (0.004) | 0.081 (0.005) | 0.171 (0.004) | 0.205 (0.004) | 0.094 (0.004) | 0.082 (0.005) |
| v724 | 0.155 (0.004) | 0.198 (0.004) | 0.081 (0.004) | 0.081 (0.004) | 0.167 (0.004) | 0.205 (0.004) | 0.091 (0.004) | 0.082 (0.004) |

A Taylor diagram is a convenient 2D graphical representation focusing on the statistics R, σ and STDD, which are by nature debiased (mean-subtracted). The so-called 3D version given in Figure 22 makes the bias information available as a colour scale. Such a diagram is a polar coordinate representation of (σ, R). The standard deviations of series σ are used as the radius, and the correlation, R, with respect to a common reference is converted into an angle using acos(R). It is worth noting that the relation between the correlation and angle is highly non-linear; a 45° angle is already a 0.7 correlation. The reference data is always located at the x axis (correlation 1 with itself) and with a white marker (0 bias with itself) at the position $\sigma_I$, the reference being the in-situ data.

Figure 22 shows the performance of the overall retrieved soil moisture time series obtained from L1C V620 (1) and from L1C V724 (2) against the reference in-situ time series (0). Thanks to the concatenation, a large number of points (~40,000) allow computing reliable statistics, which result in a narrow CI95 that does not overlap for R, bias and STD making the separation of plots significant.

Compared to v620, v724 increases the correlation with respect to in-situ data and obtains an $\sigma_S$ closer to $\sigma_I$. As usual, this is more prominent for ascending morning orbits, where Level 2 retrievals always perform better, with better thermodynamic equilibrium at the surface and a calmer ionosphere in the mornings than in the evenings. Different RFI contamination patterns are also likely playing a role.

These two results indicate an increase in signal-to-noise ratio for v724, generating less noisy retrieved soil moisture and possibly better long-term stability. However, for the latter, two years of data is probably too short, and it is necessary to wait for the full $L_1$ and $L_2$ 10-year reprocessed data availability. Finally, similarly to the spatial maps, using the v724 data, provides here ~1% more successful retrievals in these time series.

## 4. Conclusions

The SMOS team has started a new reprocessing campaign, the third, after several improvements have been introduced in calibration and image reconstruction. In calibration, the changes mainly affect the NIR calibration parameters, the NIR antenna losses, the PMS sensitivities and the correction of the thermal coupling in one important thermistor. In image reconstruction, the changes focus on reducing the spatial biases induced by the dissimilarities of the antenna patterns, and on reducing the Sun effects in the image, which cannot be considered as a point source at L-band. These corrections improve the quality of the data, as indicated by several metrics that analyse spatial biases, measurement stability, and other image reconstruction errors, as well as by comparisons against in-situ measurements and $\chi^2$

metrics from soil moisture retrievals. This reprocessing campaign comes just after SMOS has been in orbit for over 10 years.

**Author Contributions:** NIR calibration, R.O. and I.C.; Antenna losses, I.C., J.K. and R.O., PMS Sensitivities, J.C., I.C. and A.Z.; PMS Heater correction, J.C., I.C. and A.Z.; Antenna patch thermistor correction, J.K. and R.O.; Gibbs-2, A.K. and F.C.; Ice-sea mask, F.C.; Super-sample Sun correction, F.C.; Sun correction in the back, A.K., Software, J.B. and G.L.; Supervision, M.M.-N., R.O. and R.C.; Validation, J.T., P.R., R.O., R.D.-G., V.G.-G.; Writing—original draft, R.O., V.G.-G.; Writing—review & editing, I.C., J.K., J.C., P.R., J.T., J.B., G.L., R.C., M.M.-N., A.K., V.G.-G. All authors have read and agreed to the published version of the manuscript.

**Funding:** This work has been funded by the European Space Agency under the SMOS programme.

**Conflicts of Interest:** The authors declare no conflict of interest.

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
