# Peer review of "SMOS Third Mission Reprocessing after 10 Years in Orbit"

_remotesensing, doi:10.3390/rs12101645_

Round 1

Reviewer 1 Report

The authors introduced the new SMOS 3rd mission reprocessing (v724) and focused on the effects of processors changes on SM and VOD retrievals by reference to the current baseline processor v620. The paper was well organized and the point-by-point descriptions were clear. However, numbering of Figures should be rechecked (e.g., Figure 1 in Line 530?), and some abbreviations should be given full names at their first appearances (e.g., NIR, I thought it was near-infrared at my first glance).

I wonder how SM and VOD have been changed so much in Figure 18. A large global area has negative SM biases as much as -0.05 m3/m3. Note that the current SMOS-L3 and SMOS-IC products already underestimate SM over moderate and dense vegetation. Although you reported slightly wetter biases relative to the processor v620, most of your validation sites were distributed over areas of moderate changes. If you incorporate more sites in the western USA and north Africa, large wet biases might be derived.

I wonder whether SM/VOD retrievals are so sensitive to BT changes or the BT changes in the processor v724 are so large.

Author Response

We thanks the reviewer for the comments he raised.

Abbreviations and Figure references have been revised in the new manuscript

Indeed, it is the second time in all the L1 versions evolution that a L1C version is different enough to report noticeable differences at retrieved SM and VOD level other than variability  around a zero mean. The last time we observed significant changes (improvements) perceptible at L2 SM level was with the v505, 7 years ago. 

Most of the SM/VOD changes  appear also at specific locations where actually the brightness temperatures level and/or polarization directional profiles have necessarily changed enough. There is also connection with relatively wet and high VOD along also contrasted scene having large surface contrast either ocean/land or land/forest.

From our understanding, we can think of three elements that possibly play a role on these changes:

1) L1 metrics over Ocean: most of the L1 metrics are derived from the Ocean surface in order to have a relative geophysical stability that land surface cannot offer. It has  a consequence, it concerns quite cold TA (~100K ) compared to land (~300K). All effects scale with TA and we certainly expect larger variations over the much warmer land surface than those reported for the Ocean surface.

2) The Gibbs 2 effect correction: for these area where strong temperatures contrast is in the front of the instrument, Gibss2 is more prone of the strongest brightness temperatures changes. Over the warm land surface it can reach 2 to 3 time the RA with brightness temperatures angular shape change=> values ~5K to ~15K (see for instance Khazaal 2019)

3) Non linear brightness temperatures dependency to SM,VOD: finally for these surfaces either relatively wet (SM > 0.2) and/or dense vegetation (VOD > 0.3), it is also where the sensitivity of brightness temperature to SM and VOD has already started to to decrease significantly. For a given change in brightness temperatures the variations of SM and VOD need to be larger to catch up compared with surfaces having lower SM and lower VOD that report less change if Figure 18.

In summary all together: much warmer TA scenes over land than other ocean in association with Gibbs 2 correction may imply not so small changes on land BT along with medium to low sensibility for SM and VOD for these specific area can certainly generate significant enough variations on SM and VOD.

The focus of this publication is about the L1 changes. The SM/VOD section reports the final impact. Comparing coarse resolution retrieval against in-situ measurement is subject to many difficulties such as representativity, spatial scale effects, retrieval errors, in-situ errors, observation errors (RFI). The goal was to assess the relative changes between the v620 and v724 and not to provide a SM/VOD validation difficult exercise which is out of scope.  The above difficulties become less an issue when it turns to do a differential assessment as all the enumerated limitations are the same for the v620 and v724 versions.

Regarding the validations sites we try to keep the list of them stable between successive L1 evolution in order to keep an identical reference and thus comparable L2 SM / L1C metrics. The set we considered is a trade-off between the fact that we need long enough time series, 2 years is a minimum, the capability to process them in reasonable time (10 days) and the will to cover as many different surfaces type we can. So it was not possible for such exercise to include all known sites e.g. from ISMN. The North American sites (SCAN, MESONET and WATERSHED) represent already ~70% of the sites we consider, half of them being western.

It is certainly true that other sites we do not included might have retrieved SM than in-situ measurements and so with potential associated wet bias instead of the overall increased negative one. We believe that it would probably not change the conclusion that the v724 generate on overall sightly lower SM and marginally significantly lower SM compared to v620. It would be favorable to in-situ sites where the retrievals are wetter and detrimental to sites where the retrievals are drier.

However, though it is an interesting evaluation and discussion, it would be more appropriate for a future SMOS validation paper focusing on Soil Moisture retrieval, once the full new L1 v724 + new L2SM v700 reprocessing campaign will be over after this summer

Reviewer 2 Report

The paper is well-written and for sure is suitable for readers of Remote Sensing journal. Although the methodology section is too technical, but when going through the Result section, it become more clear. My minor suggestion and comments are as follow:

Line 68: Please define Tna and Tnr here.

It is recommended to present (maybe as a supplementary section) similar figures to figures 18 and 19, but for a winter month, e.g. December or January to see the performance of v724 against v620.

Line 444: something have gone wrong here, please check.

Line 512: With regard to the Result section, e.g. Figures 16,17, 18 and 21, we can conclude that v724 has been improved but not substantially improvement has been happened.

Author Response

We thank the reviewer for his answers.

Tna and Tnr have been defined in the new revised version

We have added a sentence mentioning that similar plots are obtained in different months, but considered that the paper is large enough as is, and prefer not to include more figures

Reviewer 3 Report

The manuscript entitled "SMOS 3 rd Mission Reprocessing after its 10 years in-orbit" by Oliva et al presents the revised processor developed to reprocess 10 years of SMOS measurements and its impact in term of quality of primary measurements and on estimated surface soil moisture and VOD. The study is of interest for the readers of Remote Sensing, especially for those interested  in the estimation of VOD and soil moisture from passive sensors, as it sheds light quite clearly on the whole process (and associated uncertainties) leading to the final estimations and products. The description is well and clearly written, and illustrated in a good-balanced way. Even if the manuscript is of great interest, I personally have a doubt on in which category the manuscript should be. If it is a Research Paper, I think the presentation should have been different: first a description of what is missing to previous research (errors in SSM and VOD estimations due to limitations from SMOS previous processor, but also from similar chains, i.e. more general than SMOS mission), then description of the methods to tackle them, with appropriate mentions on how innovative they are and then the results with the recommendations, and a view of its implementation. In the present form, the manuscript, though interesting, may to my view be qualified differently, as a Technical Note, or some kind of Review paper, as it refers directly at the technical description of the processor, followed by a short impact study. I insist on the fact I find the manuscript well documented and potentially very interesting for the readers.

Would it be possible to clarify that? I guess it is a question of reformulation in case of research paper (with some more context and references to other missions, more justifications at the beginning), or a change from Research paper to another qualification available in Remote Sensing, which would allow such kind of useful, but more technical documentation.

2 more minor comments:

  • A diagram of the processor may be nice for the readers.
  • l 444: check the pointed reference

Author Response

We thank the reviewer for the time and dedication to review our paper.

We appreciate very much his comments about the fact that the manuscript is well documented and very interesting for the readers. We agree that this manuscript will be very valuable for all SMOS data users.

Regarding the category of this publication, we still believe that it should be considered as an Article. The purpose of this publication is to focus on the changes in calibration and image reconstruction from the L1 processor, not on the Soil Moisture validation. As such, each section of the calibration and image reconstruction starts describing the problems observed and then describe the proposed algorithm or characterisation changes and finally the results for that particular modification, as in the proposed restructuration by this reviewer.  In the end, the Results section provides an end-to-end evaluation of all modifications combined.  We agree that this publication differs slightly from the traditional research paper because there are many changes applied, but we believe that grouping all these changes in one single article is extremely useful for the SMOS users.

However, we will also accept a possible change of article type denomination if the journal editor believes is necessary.

We have also included the minor modifications requested

Round 2

Reviewer 3 Report

I would like to acknowledge the additions provided by the authors. I still maintain the manuscript is of interest for all the users of SMOS derived products, including soil moisture. And I still find the paper is not exactly a "research paper", for the reasons invoked previously. But, after browsing through the types of publications in Remote Sensing, it seems there is no class that fits exactly... I would suggest the editors to consider to publish it as a research paper, but maybe consider the possibility in the future to broaden the scope of existing categories (such as "technical" papers or "reviews") to such kind of paper.